# A Flexible Method for Nanofiber-based 3D Microfluidic Device Fabrication for Water Quality Monitoring

**DOI:** 10.3390/mi11030276

**Published:** 2020-03-06

**Authors:** Xiaojun Chen, Deyun Mo, Manfeng Gong

**Affiliations:** School of Mechanical and Electronic Engineering, Lingnan Normal University, Zhanjiang 524048, China; gongmf@lingnan.edu.cn

**Keywords:** nanofiber-based, 3D microfluidic chip, water-quality monitoring, electrostatic printing

## Abstract

Water pollution seriously affects human health. Accurate and rapid detection and timely treatment of toxic substances in water are urgently needed. A stacked multilayer electrostatic printing technique was developed for making nanofiber-based microfluidic chips for water-quality testing. Nanofiber membrane matrix structures for microfluidic devices were fabricated by electrospinning. A hydrophobic barrier was then printed through electrostatic wax printing. This process was repeatedly performed to create three-dimensional nanofiber-based microfluidic analysis devices (3D-µNMADs). Flexible printing enabled one-step fabrication without the need for additional alignment or adhesive bonding. Practical applications of 3D-µNMADs include a colorimetric platform to quantitatively detect iron ion concentrations in water. There is also great potential for personalized point-of-care testing. Overall, the devices offer simple fabrication processes, flexible prototyping, potential for mass production, and multi-material integration.

## 1. Introduction

Monitoring and control of water pollution are vital tasks worldwide. The detection of organic pesticides, phosphorus, potassium, nitrogen, and heavy metals in water is an effective way to meet drinking water standards [1]. Detection methods include spectrophotometry, atomic absorption spectrophotometry, inductively-coupled plasma atomic emission and mass spectrometry, and atomic absorption spectroscopy, which require a spectrophotometer, a microplate reader, an atomic absorption spectrophotometer, and an optical emission spectrometer [2]. These instruments have high sensitivity and low detection limits but they are expensive, complex to operate, require professional personnel, and have long detection times. Hence, their use is limited in developing countries, remote areas, point-of-care testing and individual households. Therefore, the development of rapid, low-cost detection methods is needed to analyze water quality.

Paper-based microfluidic chips (µPADs) [3], are low-cost, rapid, simple to operate, and disposable. They have broad application prospects in environmental monitoring, water-quality testing, and health care [4,5,6,7]. Whitesides et al. combined smart phones with µPADs for simultaneous detection of glucose and protein in artificial urine samples [8]. Chen et al. employed nano-gold particles on µPADs to quantitatively detect mercury ions in water [9]. Zheng et al. produced three-dimensional (3D)-µPADs for quantitative detection of heavy metal concentrations in water samples through colorimetry [10]. Recently, µPADs have been fabricated on paper via photolithography [11], spray printing [12], paper cutting [13,14], inkjet printing [15,16], and screen printing [17]. The 3D-µPADs increase chip functionality without increasing chip size. Currently, stacking 2D layers [18,19] and paper origami [20,21] have been used to fabricate 3D-µPADs. The first technique involves the combination of multi-layer paper stacked together using patterned paper and double-sided adhesive tape (or hydrophilic spray adhesive). Holes are filled with cellulose powders to provide vertical connections between adjacent layers [22,23]. This bonding process is labor-intensive and complicated. Accordingly, the potential issues include misalignment of paper/tape layers and discontinuity of channels in adjacent paper layers. As an alternative technique, origami involves layering or folding patterned paper into a 3D stack of μPADs without extra bonding steps. However, this folding-based approach still requires the use of a pre-fabricated clamp (or lamination) to hold the stack together. Therefore, a method for directly manufacturing 3D-µPADs is of great interest.

Nanofibers were fabricated by continuous spraying of solutions using electrospinning technology. The electric field, collection distance, and speed of movement could be adjusted to obtain nanofiber mats of arbitrary thickness, different pores, and multiple materials [24]. It exhibits good flexibility in terms of chemical reaction time control [25,26], and fluid velocity control [27]. Compared with paper-based microfluidic devices, the fabrication of three-dimensional nanofiber-based microfluidic analysis devices (3D-µNMADs) involves the electrospinning of nanofibers in a layer-by-layer stack and electrostatic direct writing of hydrophobic barriers. It avoids additional processes, such as alignment with auxiliary fixtures and bonding with double-sided tape. Therefore, nanofiber membranes are thus a very promising alternative to paper µPADs.

A simple and flexible method for fabricating 3D-µNMADs for water quality testing using electrostatic printing technology is discussed. Nanofiber membranes and wax hydrophobic barriers are printed alternately to build 3D-µNMADs. The flexible printing capability enables one-step fabrication of 3D microfluidic devices without the need for alignment or adhesive bonding. Deposition of the nanofiber substrate is controllable, and different substrate materials can be arbitrarily combined to form multifunctional 3D-µNMADs. To demonstrate a practical application, a 3D-µNMADs colorimetric platform was fabricated to detect the iron ion concentration in water. The 3D-µNMADs show significant promise for rapid and flexible environmental monitoring, especially in resources-limited regions.

## 2. Materials and Methods

### 2.1. Materials and Solution Preparation

A polyimide (PI) and dimethylacetamide (DMAc) mixture was purchased from Hangzhou Surface Science & Technology (Hangzhou, China). Silicon powder (Si ≥ 99%, 200-mesh) was purchased from Sinopharm Chemical Reagent Co., Ltd. The silicon powder was mixed with the PI using a magnetic stirrer (JJ-1, 1500 r/min, Jiangsu, China) at room temperature for more than 24 hours until it became a homogeneous solution. Wax was obtained from China Petroleum and Chemical Corporation, Inc. (Maoming, China). Ferric chloride (FeCl_3_) and potassium ferrocyanide (K_4_Fe(CN)_6_) were purchased from West Long Chemical, Inc. (Shantou, China). The FeCl_3_ was ultrasonically dissolved in deionized water for 2 hours. The K_4_Fe(CN)_6_ was dissolved in deionized water to form a 0.15 mol/L solution and was used as a colorimetric reagent. Food dye solutions were purchased from Huizhou Fengge Food Co., Ltd. Deionized water was obtained from the Pen-Tung Sah Institute of Micro-Nano Science and Technology in Xiamen University. 

### 2.2. Fabrication of Micro/nanofiber Membranes

A schematic illustration of nanofibers prepared by electrostatic spinning is shown in Figure 1a (left), which depicts a precision syringe pump (Harvard/11 Pico Plus, Harvard Apparatus, Holliston, MA, USA), a spinneret (0.34 mm inner diameter), and a DC high-voltage source (DW-P403- 1AC, Dongwen, Tianjin, China). An aluminum foil collector was placed on top of an X-Y motion stage (Googoltech, Shenzhen, China). The anode of the high voltage source was connected to the conductive spinneret and the ground collector was connected to the cathode. The flow rate of the PI solution was 200 µL/h, and electrospinning was performed under an applied bias of 8 kV with a spinneret-to-collector distance of 5 cm. The collector was moved horizontally at 35 cm/min in the X-axial direction with a displacement of 5 cm, and then moved intermittently in the Y-axial direction for better distribution and uniformity of the nanofibers. Different thicknesses of nanofiber membranes were obtained by depositions of 2–6 h. All experiments were conducted at room temperature, 1 atmosphere pressure, and approximately 50% relative humidity. 

### 2.3. Wax Patterns by Electrostatic Printing

When the nanofibers were deposited with a predetermined thickness, the X-Y stage was moved to the position for wax pattern printing (Figure 1a, right). The wax was placed in a stainless-steel syringe (OZ-50 mL, Yanmeng Co, Ltd. Qinghuangdao, China) with temperature adjustment. The wax was melted and remained molten at 100 °C. The liquid wax was squeezed out of the nozzle using a peristaltic pump at 500 µL/h. Then, the high voltage power supply provided an electrostatic field between the nozzle and the collector. The collector was mounted on a two-dimensional platform that moved at 2.5 mm/s. The nozzle-to-substrate distance was 3 mm and the substrate on the X-Y stage moved at 0–200 mm/s. The heating temperature was read from the heating control system. At the applied high voltage, wax was ejected and deposited onto the substrate as continuous droplets in a predesigned pattern (Movie S1, Supporting Information). 

### 2.4. Fabrication of the 3D-µNMADs 

Many methods have been reported to fabricate the µPADs, such as photolithography, ink jet etching, plasma treatment, paper cutting, wax printing, inkjet printing, flexography printing, roll-to-roll thermal imprinting, and screen printing. The fundamental principle of these fabrications was to form hydrophilic–hydrophobic barriers on chromatography or filter paper to create fluidic channel networks [28]. Here, the 3D-µNMADs fabrication had two key steps. Nanofiber membranes were printed as structural materials, and then wax materials were printed as hydrophobic boundaries of the microchannels. These steps were repeatedly performed according to a predetermined computer-guided trajectory. The layered schematic of the multilayer channel structure consisted of inlet and outlet, upper channel, intermediate connection layer, and lower channel structure (Figure 1b). The fabrication process of the multilayer channel structure is shown in Figure 1c. The overall fabrication process was the alternated printing of nanofiber membranes and wax patterns. The 3D-µNMADs were then placed on a 120 °C hot plate for several seconds to allow the wax to melt and fully penetrate the patterned nanofiber membranes. This process created completely hydrophobic barriers that defined the hydrophilic channels, fluid reservoirs, and reaction zones from a simple computer design.

### 2.5. Characterization

The nanofiber membrane and wax pattern morphologies were imaged using field emission scanning electron microscopy (FESEM, SUPRA 55, Zeiss, Oberkochen, Germany) at a 15-kV acceleration voltage. An optical microscope (MF-U, Mitutoyo Measuring Instruments (Shanghai) Co., Ltd., Shanghai, China) was also used to image the wax patterns. The color intensity of the colorimetric detection zones was measured with ImageJ software (version 1.80).

## 3. Results and Discussion

### 3.1. Demonstration of the 3D-µNMADs 

The micro/nanofiber membranes were fabricated as the structural material, and wax lines or films on the membranes were used to create hydrophobic barriers in a repeated fashion, which formed the 3D multilayer channels shown in Figure 2a. The 3D microfluidic devices were tested using the capillary flow of dye solutions in the microchannels of the upper and lower structures. As shown in Figure 2b, red dye (lower channel) and blue dye (upper channel) flowed from the inlets to the outlets without clogging or leakage. Thus, the fabricated 3D-µNMADs were functional. Compared with conventional methods, the fabrication method did not require the use of double-sided adhesive tape. Previous methods [29] were time consuming, required auxiliary clamps or folding to ensure contact between layers of paper, and required precise manipulation to fill the holes in the adhesive tape. In contrast, the fabrication here was easy, flexible, fast (batch electrospinning [30,31]), and required no auxiliary devices or filler, and could be adapted to produce 3D-µNMADs at a large scale. 

### 3.2. Optimization of Electrostatic Wax Printing Parameters.

Electrostatic wax printing is a atomization process [32], whereby wax is deposited on nanofiber membranes in the form of droplets, and the size of the hydrophilic channels is controlled by adjusting the temperature. Therefore, control parameters, such as the driving voltage, the speed of the platform, the distance between the nozzle and the substrate, and the heating temperature and time, have significant effects on the stability and uniformity of wax droplets or lines. The effect of the electric field driving voltage on the Taylor cone shape was investigated. Increases in the driving voltage (4–6 kV), decreased the length of the Taylor cone from 0.88 mm to 0.5 mm (Appendix A, Supporting Information). When the length was too long, a bulk wax droplet could easily fall on the substrate material and damage the microchannel structure under the driving voltage. In addition, as the voltage increased, the droplet diameters decreased, as shown in Figure 3. At 6.5 kV, the average droplet diameter was 26 µm. The relationship between droplet diameter and ejection frequency at different voltages is plotted in Figure 3b. 

According to previous reports, liquid viscosity and surface tension decrease with temperature [33]. Therefore, the viscosity and surface tension of the wax solution are important factors in atomization and were measured at different temperatures, as shown in Appendix A (Supporting Information). When the heating temperature was 80 °C, the viscosity and surface tension of the solution were large. The stretching time required under the action of an electric field force was prolonged. Therefore, the droplet ejection frequency became slower and its diameter increased. At 100 °C, the diameter was 27 µm, while the ejection frequency increased to 205 droplets/s (Figure 4a). The frequency of the wax droplets increased with temperature, while their diameters decreased (Figure 4b). The effects of liquid viscosity and surface tension on droplet diameter were verified using the finite element simulation software (Appendix A, Supporting Information). The simulations revealed that both affected the droplet diameters and the ejection frequency. 

Because the wax droplets were directly deposited in the presence of an electric field force between the nozzle and the collection substrate, a key parameter was the nozzle to substrate collection distance. Figure 3 shows the morphologies of the wax droplets at collection distances of 3 and 20 mm. At a distance of 20 mm, the droplets were discrete and the poor deposition failed to form a continuous dense wax line. The deposition time of the wax droplets increased as the collection distance increased. Therefore, increasing the distance from the nozzle to the collection plate resulted in a more discrete droplet ejection. However, at a distance of 3 mm, the wax droplets connected to form a dense line for an improved deposition. Hence, decreased smaller collection distance improved the deposition. 

The appearance and diameters of the wax droplets were also affected by the speed of the platform (Figure 5a). If the platform was moving too slowly (less than 2.3 mm/s), the droplets accumulated together at the beginning and the charge between the droplets failed to dissipate. Hence, repulsive forces continuously acted on the subsequently printed droplets, which shifted and scattered them [34]. If the platform moved faster than 2.9 mm/s, the wax droplets were too dispersed and failed to form a continuous wax line. Conversely, the line width increased with the accumulation of wax droplets. When the moving speed was 2.7 mm/s, the wax droplets were sparse although they could be connected in a line. Figure 5b shows the relationship between the speed of the platform and the line width. A minimum average line width of 42 µm could be generated by electrostatic wax printing at a collecting plate speed of 2.5 mm/s. 

The wax droplets were deposited onto the nanofiber membrane after atomization, and they partially penetrated into the membrane (Figure 6a). Molten wax was used to fill in the nanofiber membrane porosity to create hydrophobic regions. The SEM image in Figure 6c shows the morphology of the nanofiber membrane filled with molten wax, where pink food dye was dropped on a hydrophilic channel. The penetration of the dye solution in the channel revealed the boundary of the hydrophilic–hydrophobic channel, which maintained excellent integrity. This boundary would prevent liquid leakage or cross-contamination during solution transportation. 

The penetration of wax droplets into the membrane was a capillary flow that could be described by the Lucas–Washburn equation. The droplets flowed in both the lateral and vertical directions (Figure 7a). The vertical penetration formed hydrophobic barriers that controlled the channel heights (H_h_). Lateral spreading generated hydrophobic barriers that controlled the channel widths (W_h_). An important parameter was the heating time at a fixed temperature because it determined the viscosity and spreading of the molten wax [35]. The effect of heating time on H_h_ and W_h_ was analyzed from the cross-sections of the spread wax patterns (Figure 7b). As expected, the heating time showed a linear relationship with the square of the heated H_h_ and W_h_. The cross-sections revealed differences in lateral and vertical wax spreading because of the horizontal alignment of the nanofibers. 

Compared with the previous reports (Appendix A, Supporting Information), the 3D-µNMADs had outstanding advantages in the low-resolution microfluidic channels. The minimum size of the hydrophilic channels was approximately 296 µm after heating for 15 s, as shown in Figure 8. The typical resolution of hydrophilic channels in µPADs is approximately 500 µm [36]. Electrostatic wax printing could thus provide adequate resolution for the fabrication of 3D-µNMADs. The porosity and mechanical properties of the nanofiber membranes were also compared with previous reports (Appendix A, Supporting Information). The pure nanofiber membranes had a lower porosity and better elasticity than paper, and could be used for future stretchable microfluidic devices.

### 3.3. Performance of 3D-µNMADs

In Figure 9a, the capillary force fluid flow rates were plotted for the nanofiber membranes thicknesses. Specifically, 10 µL of a dye solution was pipetted into four membranes with different thicknesses. The flow rate had a linear relationship with the thickness at 35 s, and absorption by the samples increased with thickness. To demonstrate how the flow rate could be adjusted, a multi-channel mold with a round liquid storage chamber was fabricated by 3D printing, as shown in Figure 9b. There were four channels that were all 5-mm wide, 20-mm long, and had depth gradients of 45–100 µm. When dye solution was dropped in the center, the flow correlated with the membrane thickness because the distance of the flow differed in the vertical direction. The flow was faster along the length direction in the thin membranes relative to that in the thick membranes, as is predicted by the Lucas–Washburn equation. In addition, the membrane porosity decreased and became denser after a long deposition time, which resulted in a reduced capillary flow [37]. However, if the nanofiber thickness was too thin, it could break during fabrication.

In previous reports, solution inhomogeneity in the reaction zone has been shown to affect the efficiency and accuracy of the detection analysis [38]. Thus, ethanol was used to treat the µNMADs (Figure 9c). Treated nanofiber-based microfluidic analysis devices (µNMADs) had a more uniform solution distribution at the interface; however, they had slower flow rates relative to those of the untreated µNMADs. One advantage of treatment is for short-range biochemical analysis. The compactness of nanofiber membranes may affect capillary action, making the transport of fluids worse than that in filter paper. Thus, nanofiber membranes were fabricated by doping silica powder in the PI solution, which is an advantage of flexible electrospinning. The nanofiber membrane had a porous structure that increased internal fluid flow, and exhibited stronger capillary action [39,40] (Appendix A, Supporting Information). The rapid transport of fluids could allow samples or reagents to reach the detection position faster, which is conducive to saving detection time. Shou et al. presented a novel nano- or microfibrous hollow wedged channel which can significantly speed up the capillary flow [41]. It was a very simple and practical method to obtain fast capillary flow by changing the structure of the fiber. In addition, they also proposed a general quantitative model of capillary flow in a uniform porous medium with varying cross-sectional dimensions. The regulation and maximization of flow rates in porous materials is of great significance for a variety of applications including biomedical diagnostics, oil recovery, microfluidic transportation, and fabric water management [42].

### 3.4. Assay of Iron Ion Concentration in Water

Food and drinking water are the main sources of iron for the human body. Excess iron can affect the heart and could be more dangerous than cholesterol. Here, an Fe^3+^ assay was demonstrated with an eight-region (No. 1–8) colorimetric assay on 3D-µNMADs (Figure 10). An FeCl_3_ indicator solution (0.5 µL) was individually pipetted into regions No. 1–7. The concentrations ranged from 0.001 to 0.09 mol/L, while a 0.04 mol/L solution was pipetted into region No. 8. After air-drying, 10 µL of colorless K_4_Fe(CN)_6_ detection reagent was pipetted into the central zone. It reacted with the FeCl_3_according to: 4FeCl_3_ + 3K_4_Fe(CN) _6_ = Fe_4_[Fe(CN)_6_]_3_ + 12KCl. The Fe_4_[Fe(CN)_6_]_3_ product turned the detection areas blue (see Figure 10a). The analytical device was photographed after the color reaction had completely developed. The photo was then converted to grayscale with Adobe Photoshop CS6. The Fe^3+^ concentration of each sample and the corresponding grayscale data are listed in Table 1. The detection zones of No. 1–7 exhibited a linear relationship of Fe^3+^ concentration vs. corresponding gray intensity that was least-squares fit, as shown in Figure 10b. According to this fit, the Fe^3+^ concentration in region No. 8 was 0.04 mol/L with a corresponding gray intensity of 104.96. Thus, the 3D-µNMADs had an adequate capability for quantitative analysis.

## 4. Conclusions

Stacked multi-layer nanofiber-based microfluidic analysis devices were fabricated through electrostatic printing. Nanofiber membranes were used as substrates. Hydrophilic channels were electrostatically printed with wax on the substrates. The flexible printing capability enabled one-step fabrication of 3D microfluidic devices without the need for alignment or adhesive bonding. A 3D-µNMADs colorimetric platform demonstrated detection of iron ions in water. Further work will focus on functional structures using different materials to implement more applications for nanofiber membranes.

## Figures and Tables

**Figure 1 micromachines-11-00276-f001:**
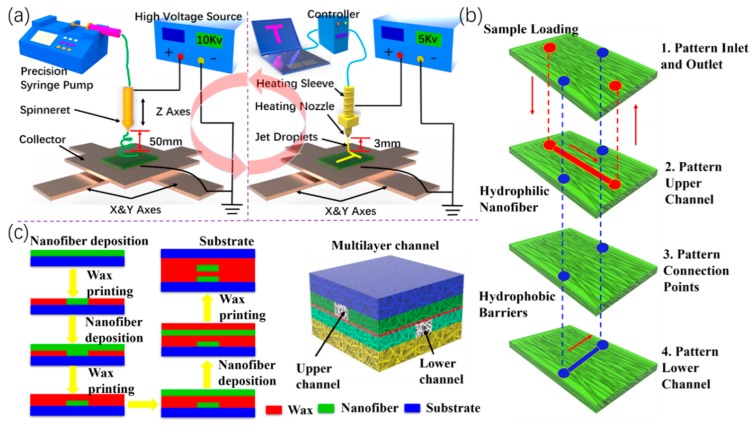
(**a**) Schematic of electrospinning and electrostatic wax printing technology. (**b**) Layered schematic of multilayer channel structure. (**c**) Fabrication process of 3D multilayer channels and final multilayer structure with upper and lower flow channels.

**Figure 2 micromachines-11-00276-f002:**
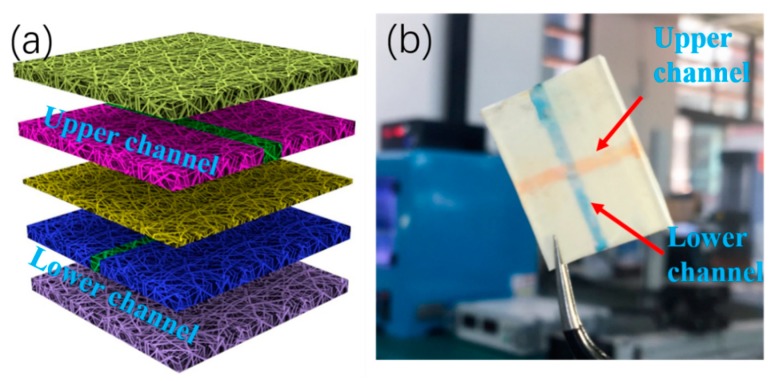
(**a**) Schematic of 3D multilayer channels. (**b**) Dye solution testing.

**Figure 3 micromachines-11-00276-f003:**
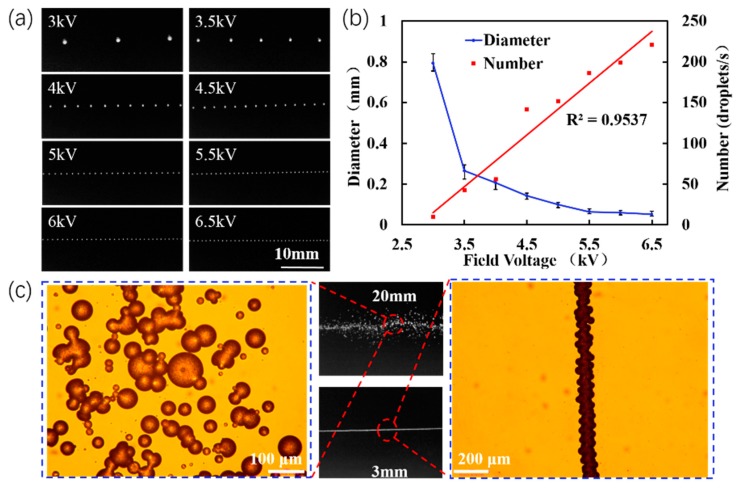
(**a**) Optical microscopy image of wax droplets formed at driving voltages of 3 to 6.5 kV. Heating temperature: 100 °C; collection distance from nozzle to substrate: 3 mm. (**b**) Droplet diameter and frequency vs. driving voltage. (**c**) Morphologies of deposited wax droplets at collection distances of 3 and 20 mm. The substrate is a silicon wafer.

**Figure 4 micromachines-11-00276-f004:**
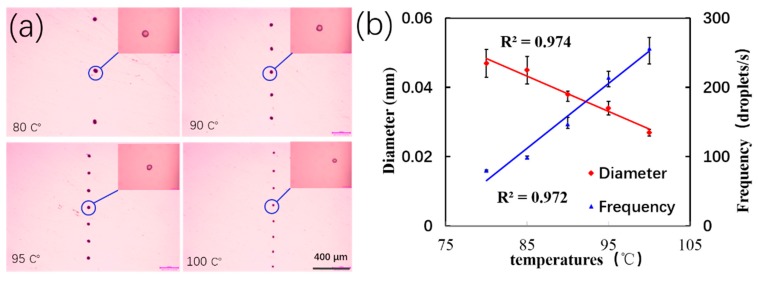
(**a**) Morphology of wax droplets at different temperatures. The background of the picture has been processed. (**b**) The relationship of diameter and frequency of wax droplets at different heating temperatures. Collection distance: 3 mm; electric field voltage: 5 kV; collecting plate moving speed: 40 mm/s.

**Figure 5 micromachines-11-00276-f005:**
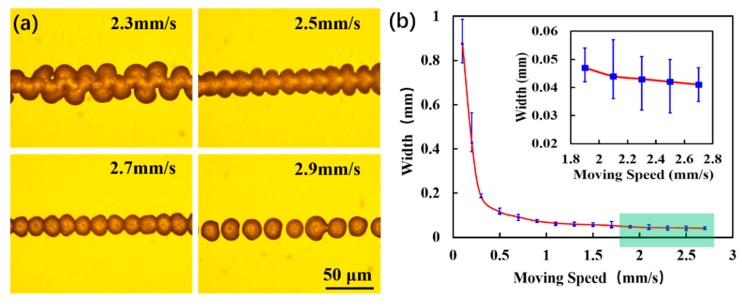
(**a**) Microscopy images of wax droplets patterns on the collecting plate at different moving speeds. Electric field voltage: 5 kV; heating temperature: 100 °C; collection distances: 3 mm. A silicon wafer is used as the printing substrate. The background of the picture has been processed. (**b**) Relationship between collecting plate moving speed and wax droplet width. The inset shows that the average minimum line width is 42 μm under continuous stable wax droplet topography.

**Figure 6 micromachines-11-00276-f006:**
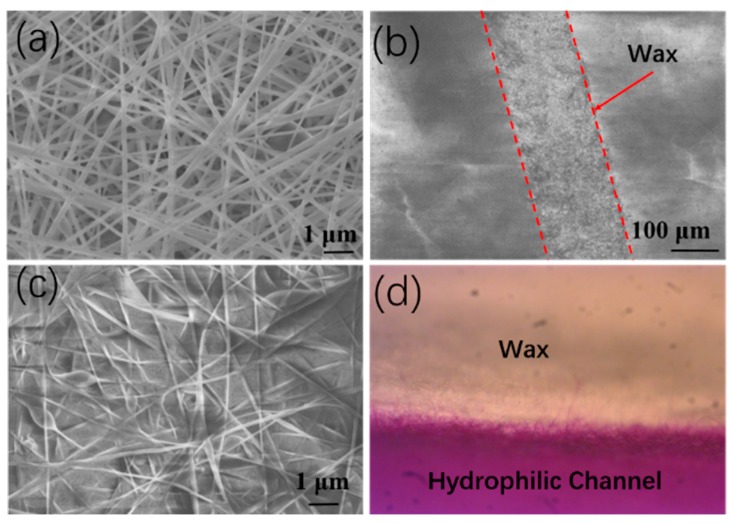
SEM (scanning electron microscopy) images of (**a**) a pure nanofiber membrane, (**b**) wax patterning in a nanofiber membrane, (**c**) wax-coated regions, and (**d**) optical microscopy images of dye transportation in the hydrophilic channel. No leakage to the hydrophobic region is observed.

**Figure 7 micromachines-11-00276-f007:**
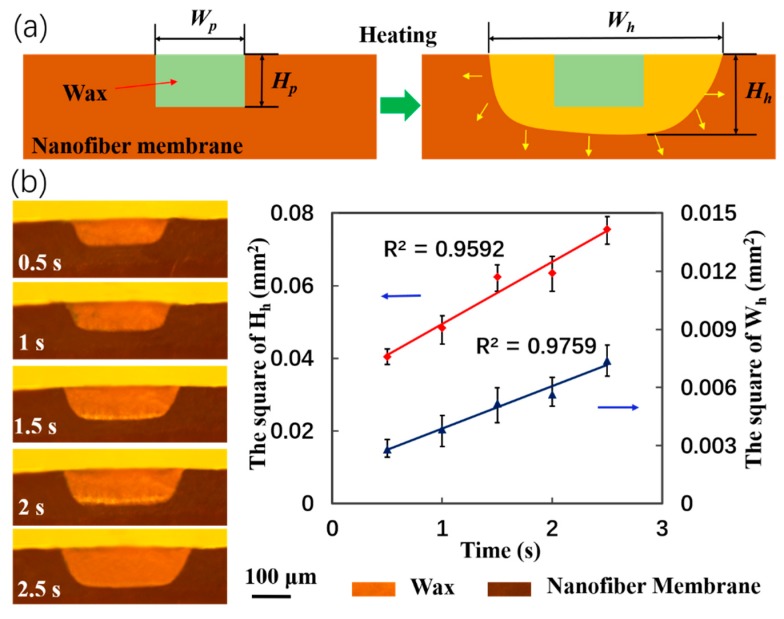
Spreading of wax in a nanofiber membrane. (**a**) Schematic cross-section of wax spreading, where W_p_ and H_p_ are the printed width and depth (green areas), respectively; and W_h_ and H_h_ are the respective values after heating (yellow and green areas). (**b**) Increasing H_h_ and W_h_ in actual images of printed wax with heating time at 120 °C. The values are averages of the barrier widths and depths.

**Figure 8 micromachines-11-00276-f008:**
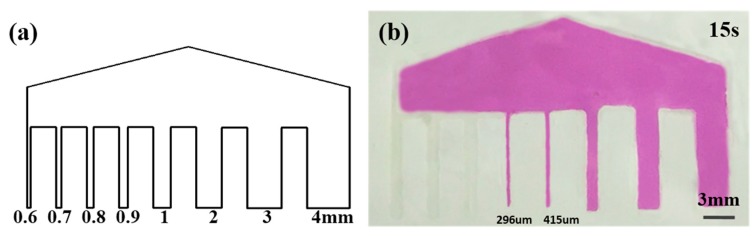
(**a**) Resolution test pattern. (**b**) Resolution of the hydrophilic channels after heating for 15 s.

**Figure 9 micromachines-11-00276-f009:**
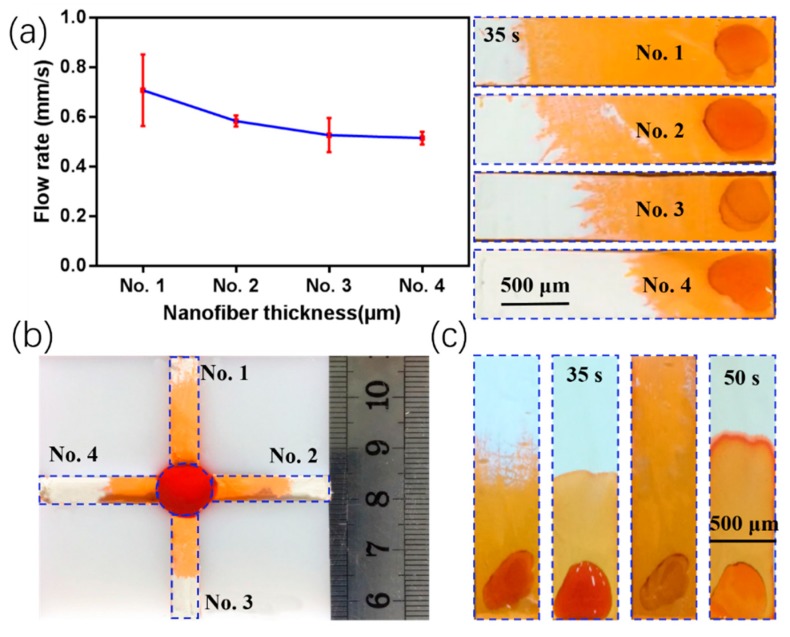
(**a**) Relationship between nanofiber membrane thickness and capillary flow rate. No. 1–4 represent four different thicknesses of the nanofiber membranes. (**b**) Image of a multi-channel analysis device that reveals the flows of yellow food dye in four channels having thicknesses corresponding to No. 1–4. (**c**) Comparison of ethanol solution treated and untreated flow conditions in nanofiber-based microfluidic analysis devices (µNMADs).

**Figure 10 micromachines-11-00276-f010:**
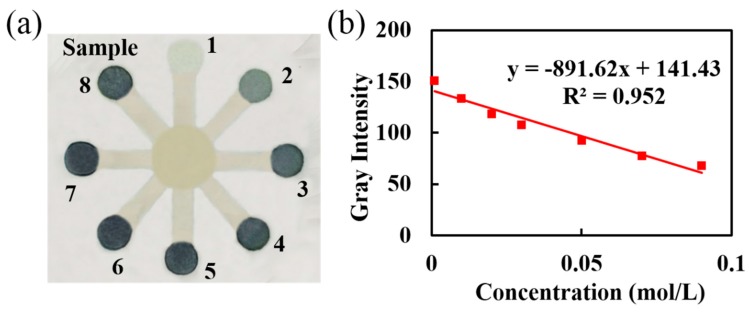
Colorimetric assay for iron ion concentration. (**a**) Image of testing regions on three-dimensional nanofiber-based microfluidic analysis devices (3D-µNMADs); (**b**) linear curve fit for gray intensity vs. iron ion concentration.

**Table 1 micromachines-11-00276-t001:** Iron ion concentration of samples 1–7 and their corresponding gray intensity.

Sample	1	2	3	4	5	6	7
**Gray Intensity**	151	135	121	108	93	77	68
**Fe^3+^ Concentration, mol/L**	0.001	0.01	0.02	0.03	0.05	0.07	0.09

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
