# Peer review of "A Flexible Method for Nanofiber-based 3D Microfluidic Device Fabrication for Water Quality Monitoring"

_micromachines, 2020, doi:10.3390/mi11030276_

Round 1
Reviewer 1 Report
The authors fabricated 3D nanofiber-based microfluidic device by programmably electrospinning nanofibers and electroprinting wax. The simple fabrication method is of interest in the field of microfluidics, lab on a chip, biomedical engineering, and environment science. The manuscript is well organized and it can be accepted after some modifications. My comments are as follows.
- It is not clear why the soft and fragile nanofibers were selected. Can the filter paper be used as the substrate?
- Will the fibrous constructs deform or collapse during heating?
- The novelty of the liquid transport in the proposed microfluidic chips has less been illustrated.
- The authors may refer to the relevant references on regulated liquid transport in fibrous media such as [Langmuir 34 (4), 2018:1235-1241] and [Physical Review E 91 (5), 2015: 053021].
Reviewer 2 Report
This article demonstrates electrostatic deposition of nanofibers and wax droplets to generate 3D patterned structures, similar to "paper microfluidics" but without the limitations associated with having to align multiple planar layers. They demonstrate with a simple colorimetric test for iron ions.
This is an interesting approach to solving the challenges with 2D paper devices, and I think will be of interest to readers of Micromachines. It can be published with only minor revisions, to address the following questions:
The tradeoff with their method, vs planar paper microfluidics, is that this is now a direct-write process. The authors should comment on how production might be scaled up beyond the research laboratory (this is the dream with paper microfluidics, that you can produce millions of devices for cheap, highly distributed diagnostics).
A fair bit of the effort is devoted to characterizing the deposition of wax in the form of droplets to ensure a continuous barrier is formed. Some of this is shown as images of droplets. I still wonder how uniform this is across an entire device (vs the small windows shown under high magnification). Some of the other images in the manuscript that show dyes localized to channels do suggest they are forming effective barriers, but I wonder if there is a way to characterize the quality of the wax layer without microscopic or visual examination, e.g. a resistance measurement.
Author Response
Please see the attachment

This manuscript is a resubmission of an earlier submission. The following is a list of the peer review reports and author responses from that submission.
Round 1
Reviewer 1 Report
The major problem for this manuscript is the conflict between the title and what is inside the manuscript. The content of the paper is devoted to describing how electro fiber spinning can be used to make a mat of spun fibers of submicron diameter and put a line of wax to bring about a hydrophobic channel. Thus it is simply a methods paper with no new information that an ordinary user of an EHD spinning machine could not do by themselves.The mat all of a sudden becomes a 3D chip!
Water quality monitoring application is minimal showing that iron could be detected colorimetrically. However, the paper does not show anything in which the microfiber based mats are superior to paper based detection technologies.
Reviewer 2 Report
This manuscript addresses a simple and flexible method for fabricating 3D-μNMADs for water quality testing using electrospinning technology. The parts of data are potentially interesting and worthy of eventual publication. However, what is proposed looks like too much a summary of technical reports. Many experimental data are only showing results, and there is almost no consideration and/or discussion. Moreover, the writing is awkward and lacks conciseness. All in all, for many reasons, this work is unsuitable for publication of “Micromachines” at this stage.
Since there are too many problems, here are some corrections.
1. Although there is a description of 2D processing method, there is no description of a multilayer method and its results. It is difficult to say “3D method and 3D device”.
2. Although there is a lot of data, details such as results and discussion are not described. Also, in the results and discussion chapter, some methods are described halfway and confused. It is very unfortunate that the value of a lot of data is lost because re-experiment is almost impossible.
3. Abbreviations should be defined in parentheses the first time they appear.
Example: Line 69, PI?
4. The description of materials and equipment is not unified.
5. The method should be written so that the experiment can be reproduced.
Example1: Line 81, PTSI-M/N Science and Technology?
Example2: Line 98, No material details of electrostatic wax.
Example3: Line 229, No material details of Liquid metal.
Example4: There is no description of the substrate used in all processing.
...
6. Numerous notation errors.
- SI Units should be used. Example: Line 91. Line 93...
- Composition notation. Example: Line 77...
- Figure # depicts “Figure X”, not “Fig.X” in “Micromachines”.
- Incomplete reference writing.
7. Many mistakes in figures
Example1: Figure 1, Separate methods and results.
Example2: Figure 1b, No description in the main text.
Example3: Figure 1d, The photo on the right does not represent the entire A-A cross-section.
Example4: There should be one blank line before and after the text and figures.
8. The description of conductivity from Line 299 seems to be important in the future, but it is one of the causes of confusion in this paper. It is better to delete it from this paper and make another one.
...
There are many other than these.
Round 2
Reviewer 1 Report
The revised version is an improvement on the original version in terms of content and suitability of the title for the work presented.
Reviewer 2 Report
The reviewer understood the authors’ intention to make corrections regarding the review comments, but it seems that the corrections are not enough. There are many comments that are the same as the first review. If the next revision can be modified to the same extent, I would like to mention reject of the paper. The authors will need to make significant configuration changes, and more caution is required.
1. Although there is a lot of data, details such as results and discussion are not described. Also, in the results and discussion chapter, some methods are described halfway and confused. It is very unfortunate that the value of a lot of data is lost because re-experiment is almost impossible. The subject of one paper should be basically limited to one point. Only the content necessary to prove the point should be included.
- There is too much supporting information. The content needed to guide the subject should be included in the main text, where methods, results, and discussions should be stated in detail.
-As pointed out in the first review, the method should be described in Chapter 2. Example: L137-L143.
-Figure 1: As pointed out in the first review, separate methods and results. In a full paper that is not conference proceedings, methods and results are not summarized in one figure. If they are put together to one figure, each description will be insufficient.
- As pointed out in the first review, the description of conductivity (integration capability, L249-L266 with Figure 7) seems to be important in the future, but it is one of the causes of confusion in this paper. It is better to delete it from this paper and make another journal paper. The above-mentioned insufficient description should be sufficiently increased using the space created.
2. Read the “Instructions for Authors” carefully and correct the entire paper.
https://www.mdpi.com/journal/micromachines/instructions
Example1: As pointed out in the first review, not only is the style of the reference list incorrect, but it is not uniform.
Example 2: As pointed out in the first review, there are still many rudimentary mistakes.
(L93: “8‐kV”. L101, L110…: Mixing tilde and dash. L116: “J apan”. ….
Round 3
Reviewer 2 Report
It is easier to understand than the original manuscript, but it is still difficult to understand due to errors and issues in the composition, and as a result, it has become clear that the novelty and usefulness of the paper is ambiguous. You should reconsider and resubmit your point.
1. Introduction
Please clearly show the novelty and the difference from the related studies shown in L45 to L50.
2. Introduction (L52)
It states that no alignment is required, but the X-Y stage plays its role, which is incorrect. Please specify the novelty and usefulness of this study. Isn't "flexible" in the title the main point?
3. Isn't L107-L120 a "method"? The method should be summarized in Chapter 2.
4. Figure 5
As a result of the experiment, could you find the conditions for forming a line with high straightness? Consideration is needed.
5. Figure 7
Are the names and units of the vertical axis of the graph correct?
6. Figure 9 (a)
The horizontal axis of the graph shows the unit of film thickness and length, but it cannot be understood from the display of No. 1-4.
7.As pointed out in the first and second reviews, there are still many rudimentary mistakes.
(Example: L276: tilde remains, L286 “NO 8” period missing, L286 “was0.04” space missing, etc.)
Editing service is required.
8. References
Due to the relevance of your paper to the journal and the convenience of readers of the journal, there should be multiple references on “Micromachines” as references.
9. Contact email address
You should refrain from using free e-mail address for contacts listed in your paper. Please use the email given by your institution.
